# Autologous Bone Marrow Mononuclear Cells (BMMC)-Associated Anti-Inflammatory Nanoparticles for Cardiac Repair after Myocardial Infarction

**DOI:** 10.3390/jfb13020059

**Published:** 2022-05-13

**Authors:** Laercio Uemura, Rossana Baggio Simeoni, Paulo André Bispo Machado Júnior, Gustavo Gavazzoni Blume, Luize Kremer Gamba, Murilo Sgarbossa Tonial, Paulo Ricardo Baggio Simeoni, Victoria Stadler Tasca Ribeiro, Rodrigo Silvestre, Katherine Athayde Teixeira de Carvalho, Marcelo Henrique Napimoga, Júlio Cesar Francisco, Luiz Cesar Guarita-Souza

**Affiliations:** 1Experimental Laboratory of Institute of Biological and Health Sciences, Pontifícia Universidade Católica do Paraná (PUCPR), 1555 Imaculada Conceição Street, Curitiba 80215-901, Brazil; l.uemura@hotmail.com (L.U.); paulo_vicmar@hotmail.com (P.A.B.M.J.); gustavoblume@gmail.com (G.G.B.); luizekremer@hotmail.com (L.K.G.); murilotonial@gmail.com (M.S.T.); prbaggiosimeoni@hotmail.com (P.R.B.S.); vicstadler@gmail.com (V.S.T.R.); julio.apfr@gmail.com (J.C.F.); guaritasouzalc@hotmail.com (L.C.G.-S.); 2Instituto de Radiologia (InRad), Hospital das Clinicas HCFMUSP, Faculdade de Medicina, Universidade de Sao Paulo, Sao Paulo 05508-070, Brazil; rodrigogms@usp.br; 3Cell Therapy and Biotechnology in Regenerative Medicine Department, The Pelé Pequeno Príncipe Institute, Child and Adolescent Health Research & Pequeno Príncipe Faculties, 1632 Silva Jardim Avenue, Curitiba 80240-020, Brazil; katherinecav@gmail.com; 4Institute and Research Center São Leopoldo Mandic, São Leopoldo Mandic, Faculty–SLMANDIC, Campinas, São Paulo 13045-775, Brazil; marcelo.napimga@gmail.com

**Keywords:** bone marrow mononuclear cells, Myocardial Infarction, rats, 15d-PGJ2, anti-inflammatory

## Abstract

To investigate the effect of transplantation of stem cells from the bone marrow mononuclear cells (BMMC) associated with 15d-PGJ2-loaded nanoparticles in a rat model of chronic MI. Chronic myocardial infarction (MI) was induced by the ligation of the left anterior descending artery in 40 male Wistar rats. After surgery, we transplanted bone marrow associated with 15d-PGJ2-loaded nanoparticle by intramyocardial injection (10^6^ cells/per injection) seven days post-MI. Myocardial infarction was confirmed by echocardiography, and histological analyses of infarct morphology, gap junctions, and angiogenesis were obtained. Our results from immunohistochemical analyses demonstrated the presence of angiogenesis identified in the transplanted region and that there was significant expression of connexin-43 gap junctions, showing a more effective electrical and mechanical integration of the host myocardium. This study suggests that the application of nanoparticle technology in the prevention and treatment of MI is an emerging field and can be a strategy for cardiac repair.

## 1. Introduction

Myocardial infarction (MI) is characterized by various etiologies, causing morbidity and mortality worldwide. After MI there is a reduced number of surviving cardiomyocytes with decrease in cardiac function and consequent remodeling of ventricle [1]. Several studies suggest that bone marrow-derived mononuclear cells (BMMC) have been able to differentiate into diverse tissues in experimental MI models, and treatment is associated with myocardial repair and functional improvement [2].

Ten years ago, short-term randomized clinical trials showed improved ventricular remodeling in patients transplanted with bone marrow-derived cells [3]. Delewi et al., 2014, reported in their meta-analysis that there was an improvement in the final volume of the left ventricle of 2.1% with (BMMC) therapy, with a small improvement in the volumes and size of the infarction, but without beneficial effects on clinical events [4,5].

However, the contribution of BMMC to myocardial repair is not yet totally understood. A promising new therapy is tissue engineering through nanomedicine with the development and application of new targeted drugs, implantable materials, and the creation of nanoscale particles for medical diagnostics and cell movement tracking [6].

The main components of nanomedicine are nanoparticles (NPS), and there are several varieties of nanoparticle types. However, we emphasize the 15-Deoxy-Δ-12,14-prostaglandin J2 (15d-PGJ2), member of a super family of nuclear receptors (PPAR-γ) agonist, which can contribute to the natural defense mechanism [7].

15d-PGJ2 nanoparticles have been widely used in various studies, because they are stable both in vivo and in vitro and easy to make. Recent studies have gained positive results using 15d-PGJ2-loaded nanocapsules (NC) to treat gout arthritis and post-infarct ventricular dysfunction in an experimental model [8,9].

In addition, PGJ has adequate qualities in regenerative medicine, such as being biocompatible as well as having low toxicity and immunogenicity [9].

Thus, in this study, we aim to assess the functional and pathological effect of transplantation of stem cells from the bone marrow associated with 15d-PGJ2-loaded nanoparticles in a rat model of chronic MI.

## 2. Materials and Methods

### 2.1. Animals

All experiments and animal studies were performed according to the guidelines for the care and use of laboratory animals by the NIH. The experimental protocol was approved by the Institutional Animal Care and Use Committee in the Pontifícia Universidade Católica do Paraná (PUCPR), Curitiba, Brazil (protocol 01248).

### 2.2. Experimental Design

Forty male Wistar rats weighing (250–300 g) were used in this study. The animals were randomized into four groups: group C (*n* = 10), MI with saline injection (control group); group (BMMC) (*n* = 10), MI with bone marrow; group NP (*n* = 10), MI with 15d-PGJ2-Nanoparticles only; and group (BMMC) + 15d-PGJ2-NPs (*n* = 10), MI with bone marrow stem cells (BMMC) + 15d-PGJ2-Nanoparticles. The animals were subjected to MI and the nanoparticle was transplanted. Thirty days after surgery all animals were analyzed by echocardiography to assess heart function. All rats were euthanized by lethal dose of thiopental sodium (100–150 mg/kg) administered intraperitoneally, and histopathological analysis was performed (Figure 1).

### 2.3. Bone Marrow BMMC Isolation

BMMC were obtained from the iliac crest of male Wistar rats (250–300 g) as previously described [10]. In short, the BMMC were aspirated into the iliac crest of each rat. The bone marrow was centrifuged at 1400 rpm for 40 min and diluted in essential Dulbecco’s modified Eagle medium (DMEM) and gently separated by the Ficoll–Hypaque density gradient method (density = 1.077 g/mL). The ring of BMMC, located in the interphase, was removed, and added to the tube, containing 20 mL of DMEM. After 2 washing steps, at 277× *g* for 10 min, the cells were resuspended in DMEM. The cells count was performed in the Neubauer chamber, and the cell viability was verified utilizing Trypan blue stain [10].

### 2.4. Flow Cytometry Analysis

Flow cytometry analysis for BMMC cellular face markers was performed on freshly isolated mononuclear cells and adherent cell populations using the FACS Calibur system (BD Biosciences, San Jose, CA, USA). Immunophenotypic analyses for CD34, CD 45, CD105, CD 90, CD73, and CD105 were performed with a commercially available kit (Stem Kit, Beckmann Coulter, Krefeld, Germany) in PBS for 30 min at 4 °C in the dark. For all the markers, immunoglobulin 1 (IgG1) was used as the isotype control [11].

### 2.5. Preparation of 15d-PGJ2 Loaded PLGA Nanoparticles

The 15d-PGJ2-NP production was prepared according to the protocol established by Alves et al. [11]. The PLGA Polymer (100 mg) was dissolved into 30 mL acetone with 15d-PGJ2 (100 μg; Sigma-Aldrich, St. Louis, MO, USA), sorbitan monostearate (40 mg), and caprylic/capric acid triglyceride (200 mg) the organic phase. Then, polysorbate 80 (60 mg) and deionized water (30 mL) were added, comprising the aqueous phase. Briefly, after the components in both phases were dissolved, the organic phase was gently added to the aqueous phase, and the suspension was maintained under agitation for 10 min. The solvent acetone (60/40 *v*/*v*). was removed under evaporation, and the suspension was concentrated to a volume of 10 mL under low pressure, using a rotary evaporator, to obtain a suspension of 15d-PGJ2 with a final concentration of 10 μg/mL. Residual solvent was removed by evaporation until there were no observed traces of acetone in the preparation and analyzed by HPLC, as previously described (Napimoga et al., 2012) [9].

### 2.6. Chronic Myocardial Infarction (MI) and Cell + Nanoparticle Transplantation

The chronic myocardial infarction (MI) model was performed according to Guarita-Souza et al. [12]. MI surgery was performed by ligation of the left anterior descending coronary artery with a 7–0 prolene suture (Figure 2).

Wistar male rats (250–300 g) were randomly assigned to four groups (*n* = 10/group) as follows: (a) myocardial infarction with saline injection (Control group); (c) (BMMC) transplantation (BMMC) group); (d) (BMMC) + NP group); and (e) 15d-PGJ2 -NP (NP group). The rats were anesthetized with 50 mg/kg ketamine and 10 mg/kg xylazine, intubated via the trachea with an 18-gauge intravenous catheter, and submitted to left thoracotomy. They were later mechanically ventilated, and the left anterior descending coronary artery (LAD) was ligated with 7.0 polypropylene thread (Ethicon^®^, Inc., Somerville, NJ, USA). For more details about the MI induction, the procedure was previously described by Francisco et al. [8].

Seven days post-myocardial infarction, the animals were anesthetized again with ketamine and xylazine before undergoing a second surgery for transepicardial with cells + nanoparticle transplantations. We applied 3 × 10^6^ (*n* = 8) (BMMC) cells and 15d-PGJ2-NP at a concentration of approximately 10 μg/mL. Each rat received three transepicardial injections (25 µL each). Each injection was given along the border zone of the infarcted myocardium scar. The protocol for cell implantation has been previously detailed [8].

### 2.7. Echocardiography Analysis

The echocardiographic analyses in rats were performed at baseline before MI, seven days after the infarct, and 30 days after cell transplantation. Briefly, animals were lightly anesthetized using 50 mg/kg ketamine and 10 mg/kg xylazine intraperitoneally and scanned by transthoracic echocardiography (Hewlett Packard Sonos model 5500) using a S12 (5–10 MHz) sectorial probe. The endpoints acquired comprised final systolic and diastolic surfaces, final diastolic and systolic lengths of the left ventricle, and heart rate to calculate the final systolic volumes (LVESV, mL), diastolic volumes (LVEDV, mL), and the left ventricle ejection fraction (LVEF %) using Simpson’s method. The echocardiography analyses were performed as previously described. [12]. This study included only animals with left ventricular ejection fractions (EF) of less than 40% of EF. The animals received perioperative analgesia, which was achieved using buprenorphine (0.5 mg/kg) and carprofen (5 mg/kg). All procedures and analyses were performed by the same researcher, who was blinded to the experimental groups.

### 2.8. Euthanasia

All animals were euthanized with a lethal dose of pentobarbital sodium (thiopental) 100 to 250 mg/kg, injected intraperitoneally.

### 2.9. Histopathological Analysis

For histology and immunohistochemistry (IHC), hearts were prepared according to the process described by Liu X. [12]. Briefly, heart tissues were fixed in 10% phosphate-buffered paraformaldehyde and were paraffin blocks sectioned into (5 μm) slices and stained with hematoxylin and eosin (H&E) to assess the myocardial structure. Gomori’s trichrome staining and Picrosirius Red were performed to assess cardiac fibrosis in the remote myocardium. The immunohistochemistry sections were incubated with antibodies specific for the detection of the following proteins: von Willebrand Factor (vWF) (ab171825, Abcam), Anti-TGF beta 1 antibody (EPR21143, Abcam), anti-cx43 (ab11370, Abcam), and anti-Desmin (ab8592; Abcam, Cambridge, UK). The next day, after washing with PBS, the sections were incubated with anti-mouse IgG secondary antibodies for 1 h in the dark. The sections were counterstained using a Horseradish peroxidase (HRP) conjugate and mounted in an antifade mountant. Heart sections stained with non-immune serum or IgG instead of the primary antibodies were used as a negative control. The data were collected from 10 individual views per heart at a magnification of ×200. The images were processed by a Zeiss Axiovert S100 TV microscope (Zeiss, Jena, Germany). Digital images were analyzed using Image-Pro Plus version 4.5 software (Media Cybernetics, Silver Spring, MD, USA). The data of the immunopositivity marking area were generated and subsequently exported to an Excel spreadsheet.

### 2.10. Statistical Analysis

The statistical analysis was performed by commercially available software (Graphpad Prism 9.1 software (Graphpad, San Diego, CA, USA). All results were expressed as the mean ± standard deviation in Tables or standard error of the mean in Figures, and differences were statistically analyzed by one-way or two-way analysis of variance (ANOVA), followed by post hoc Bonferroni’s multiple comparison tests. *p* values < 0.05 were considered statistically significant. Data were analyzed with the software.

## 3. Results

### 3.1. Characterization of BMMC

We performed flow cytometry to characterize the BMMC population using a previously published protocol [13]. Flow cytometry confirmed their mesenchymal origin; the hematopoietic stem cell markers CD34 and CD45 were negative for both these markers, whereas mesenchymal stem cell markers CD105, CD90, CD73, CD105 exhibited high expression levels. Thus, we confirmed that the major population of adherent cells was BMMC (Figure 3).

**Figure 3 jfb-13-00059-f003:**
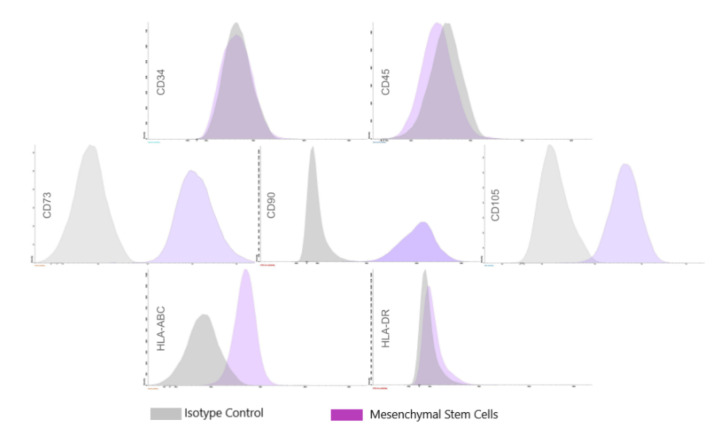
Characteristics of BMMC. Flow cytometry Histograms of Bone marrow mononuclear cells. The histograms show that the population is positive for CD73, CD90, CD105, and HLA-ABC and negative for CD34, CD45, and HLA-DR.

### 3.2. Echocardiography Findings

#### Combined Nanoparticle and BMSC Transplantation Improved Post-MI Cardiac Function

##### Intra-Group Assessment

There was no difference in LVEF on day 7 post MI (day 7) (LVEF 39%) compared to day 30 post MI (day 30) (LVEF 39.6%) in the Control group, with *p* = 0.856. There was a significant improvement in LVEF in the BMMC group in the same interval (LVEF 29.2% to 46.4%) *p* < 0.001, in the NP group (LVEF 34.4% to 51%) *p* = 0.007 and in the BMMC group + NP (39% to 51.4%) *p* = 0.011 (Figure 4).

Regarding LVESV, there was no significant difference between pre-treatment (day 7) and post-treatment (day 30) in the Control group (0.171 mL to 0.139 mL), with *p* = 0.158, and in the BMMC + NP group (0.134 mL to 0.089 mL), with *p* = 0.058. In the BMMC group there was a difference (0.233 mL to 0.130 mL), with *p* = 0.003, and in the NP group (0.176 mL to 0.088 mL), with *p* = 0.001.

In the LVEDV analysis, there was a significant difference in the comparison of the pre- and post-treatment period in the BMMC group (0.326 mL to 0.229 mL) and in the NP group (0.267 mL to 0.172 mL), with *p* = 0.008 and *p* = 0.002, respectively. There was no difference in the evaluation of the Control group (0.278 mL to 0.221 mL), with *p* = 0.077, and the BMMC + NP group (0.217 mL to 0.178 mL), with *p* < 0.001.

**Figure 4 jfb-13-00059-f004:**
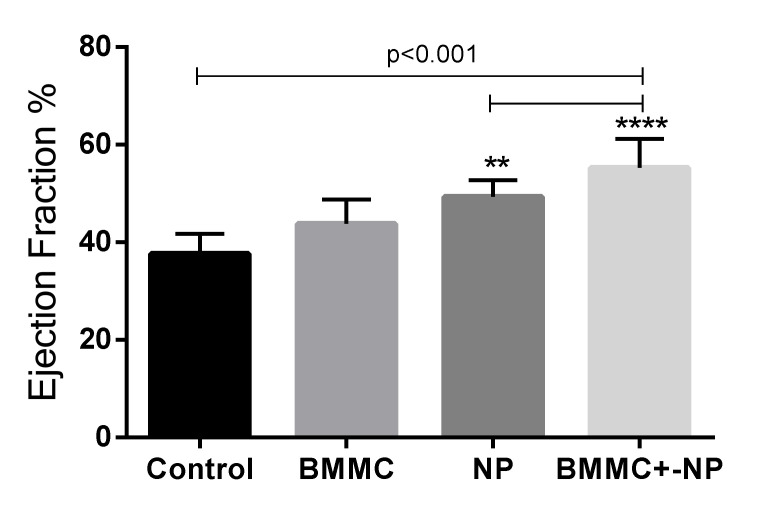
30th day LVEF: LVEF after 30 days of coronary occlusion; LVEF: ejection fraction; LVESV: left ventricular end-systolic volume; LVEDV: left ventricular end-diastolic volume; NP Nanoparticle; BMMC: bone marrow mononuclear stem cells; Values of ** *p* < 0.001 denote statistical significance.

##### Inter-Group Evaluation

The intergroup evaluation shows a significant difference in the pre-treatment LVEF in the Control group compared with the BMMC group with *p* = 0.005, and in the BMMC group compared with the BMMC + NP group with *p* = 0.007. In the other comparisons— Control group × NP, Control group × BMMC + NP group, Stem cell group × NP group, and NP group × BMMC + NP—there was no difference in pre-treatment LVEF. Regarding post-treatment LVEF (day 30), there was a significant difference when comparing the Control group with the NP group, with *p* = 0.007 favorable to the NP group (Control group LVEF 39.6%—NP group LVEF 51%). Also, after treatment, the BMMC + NP group compared to the Control group showed a significant difference, with *p* = 0.004 favorable to the BMMC + NP group (Control group LVEF 39.6%—BMMC group + NP LVEF 51.4%). In the other comparisons, BMMC × NP, BMMC × BMMC + NP, and NP × BMMC + NP, there was no significant difference (*p* = 0.231, *p* = 0.171, *p* = 0.915 respectively). In the Control group × BMMC group, there was also no significant difference in LVEF (*p* = 0.054). (Table 1 and Table 2).

**Table 1 jfb-13-00059-t001:** Echocardiographic findings in animals after chronic myocardial infarction. *p* * group comparisons at day 30.

Echocardiographic Parameters	Day	*n*	Control	Nanoparticle (NP)	BMMC	BMMC + NP	*p* *
LVEDV (mL)	7	10	0.278 ± 0.062	0.267 ± 0.053	0.326 ± 0.069	0.217 ± 0.040	0.248
	30	8	0.221 ± 0.074	0.172 ± 0.034	0.229 ± 0.071	0.178 ± 0.050	
LVESV (mL)	7	8	0.171 ± 0.053	0.176 ± 0.042	0.233 ± 0.020	0.134 ± 0.031	0.248
	30	8	0.139 ± 0.058	0.088 ± 0.036	0.130 ± 0.060	0.089 ± 0.035	
LVEF (%)	7	8	39.0 ± 7.4	34.4 ± 3.9	29.2 ± 7.7	39.0 ± 6.5	0.008
	30	8	39.6 ± 6.7	51.0 ± 10.7	46.4 ± 7.0	51.4 ± 5.4	

7th day: echocardiographic evaluation before the amniotic membrane and stem cells implantation; 30th day: echocardiographic evaluation after 30 days of coronary occlusion; LVEF: ejection fraction; LVESV: left ventricular end-systolic volume; LVEDV: left ventricular end-diastolic volume; NP: nanoparticle; BMMC: bone marrow mononuclear stem cells. Data are shown as mean ± standard deviation. Values of *p* < 0.05 denote statistical significance.

**Table 2 jfb-13-00059-t002:** Difference between pre-treatment LVEF and post treatment LVEF—inter-group evaluation.

	LVEF-Pre	LVEF-Post	LVEF-Difference
Groups Comparison	*p*	*p*	*p*
Control × BMMC	0.005	0.054	0.054
Control × NP	0.223	0.007	0.007
Control × BMMC + NP	0.985	0.004	0.004
BMMC × NP	0.145	0.231	0.231
BMMC × BMMC + NP	0.007	0.171	0.171
NP × BMMC + NP	0.243	0.915	0.915

30th day LVEF: LVEF after 30 days of coronary occlusion; LVEF:ejection fraction; LVESV: left ventricular end-systolic volume; LVEDV: left ventricular end-diastolic volume; NP: nanoparticle; BMMC: bone marrow mononuclear stem cells; Values of *p* < 0.05 denote statistical significance.

### 3.3. Histological Analyses

Histopathological analysis carried out 30 days after surgery by microscopic examination indicated that the BMMC group and the BMMC + NP group revealed the presence of factor VIII-confirmed neovascularization compared to the NP group and the Control group.

In the evaluation of Desmin, there is greater expression with a significant difference when comparing the NP group × Control group (*p* < 0.001), BMMC group + NP × Control group (*p* = 0.017), and NP group × BMMC group (*p* = 0.033). This was not observed when comparing Control group × BMMC group (*p* = 0.151), BMMC group + NP × BMMC group (*p* = 1). When Connexin-43 was evaluated to identify GAP junction, a lower expression of Connexin-43 with a significant difference was observed when comparing the NP group × Control group (*p* = 0.009), NP group × BMMC group (*p* < 0.001), and BMMC + NP group × BMMC group (*p* = 0.016). No difference was shown when comparing Control group × BMMC group (*p* = 1), BMMC group + NP × Control group (*p* = 0.445), and NP group × BMMC group + NP (*p* = 0.995).

As for the expression of TGF-β1, a difference when comparing the NP group x, the Control group (*p* < 0.001), BMMC + NP group × Control group (*p* = 0.003), and NP group × BMMC group (*p* = 0.012) was observed. No difference was found when comparing the Control group × BMMC group (*p* = 0.901), BMMC + NP group × BMMC group (*p* = 0.226), and NP group × BMMC + NP group (*p* = 1). (Table 3 and Figure 5).

**Table 3 jfb-13-00059-t003:** Intergroup analysis of the evaluation of Conexin-43, Desmin Factor VIII and TGFβ in the infarct area 30 days after MI. BMMC: bone marrow mononuclear stem cells; BMMC +15d-PGJ2-NP, bone marrow mononuclear stem cells + nanoparticle.

Variable	Group	Mean	Median	SD	*p* *
Connexin (%)	Control (*n* = 10)	9.51	9.39	0.73	0.034
	BMMC (*n* = 8)	12.44	12.04	2.03	
	Nanoparticle (*n* = 8)	12.27	12.28	0.62	
	(BMMC) + NP (*n* = 8)	6.63	6.48	1.04	
Desmin (%)	Control (*n* = 10)	4.41	3.61	1.98	0.015
	BMMC (*n* = 8)	25.01	26.60	8.19	
	Nanoparticle (*n* = 8)	17.74	18.41	4.12	
	(BMMC) + NP (*n* = 8)	6.63	6.48	1.04	
Factor VIII (%)	Control (*n* = 10)	3.23	3.47	0.71	0.018
	BMMC (*n* = 8)	9.95	7.23	6.04	
	Nanoparticle (*n* = 8)	6.63	6.48	1.04	
	(BMMC) + NP (*n* = 8)	6.63	6.48	1.04	
TGFβ (%)	Control (*n* = 10)	2.81	2.55	1.11	<0.0001
	BMMC (*n* = 8)	6.04	5.80	2.47	
	Nanoparticle (*n* = 8)	16.03	4.63	4.06	
	(BMMC) + NP (*n* = 8)	12.72	12.71	3.74	

Data are expressed as mean ± SD. BMMC +15d-PGJ2-NP, bone marrow mononuclear stem cells + nanoparticle; Statistical analysis was performed with one-way ANOVA followed by the Kruskal–Wallis test; Values of * *p* < 0.0001 denote statistical significance.

**Figure 5 jfb-13-00059-f005:**
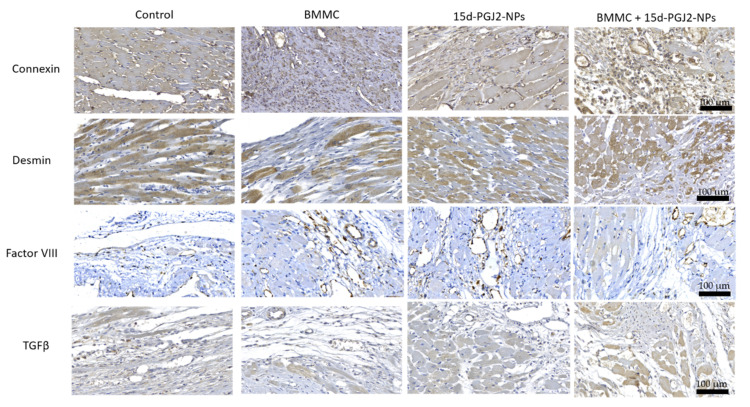
Representative immunohistochemical staining images of Connexin, Desmin, and Factor VIII; Connexin, Desmin, and TGFβ-1 at 30 days post MI (×10). Scale bar = 100 μm.

### 3.4. Collagen Level Analysis and Infarct Area

To establish the evolution of cell repair, collagen types I and III, which are the main types of collagens in healthy organisms, and the percentage of collagen types I and III in the scar area were measured.

There is a significant increase in collagen III in the Nanoparticle group, (BMMC) cells group and in the (BMMC) cells +15d-PGJ2-NP group compared to group Control I, and a significant decrease in collagen I in the (BMMC) cells +15d-PGJ2-NP group compared to group C, with statistical significance (*p* < 0.0002). The results suggested a capacity of tissue regeneration, by promoting adequate scar formation via regulating the functions of cardiac fibroblasts, was related to better cardiac function (Figure 6). There is a significant difference when comparing group Control with group BMMC) cells +15d-PGJ2-NP in collagens I and III, with statistical significance (*p* < 0.05). Cross-sections from the Nanoparticle group, (BMMC) cells group, (BMMC) cells +15d-PGJ2-NP group, and control animals were stained with Masson’s trichrome and used to quantify the infarcted area in each animal. At four weeks post-infarction, there was no significant decrease in infarct size between the four groups after 30 days of analysis (*p* = 0.979) (Table 4).

**Figure 6 jfb-13-00059-f006:**
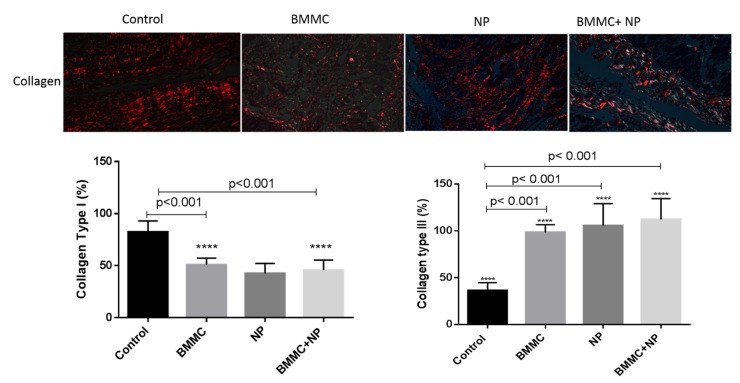
Images from Sirius red staining were obtained with circularly polarized light. Collagen type I (red fibers) was more predominant in the scar from animals treated with BMMC + NPJ than with collagen type III (green fibers). Scale bars, 50 µm. Collagen type I content index. Data are expressed as the mean ± standard error of the mean; *p* < 0.001 denoted statistical significance in comparison to control group; post-hoc comparison tests: **** *p* < 0.001 vs. Control, **** *p* < 0.001 vs. BMMC + NPJ.

**Table 4 jfb-13-00059-t004:** Intergroup analysis of infarct area and collagen types after 30 days post-infarction.

Variable	Group	Mean	SD	*p* *
Type I	Control (*n* = 10)	82.50	±10.74	<0.0001
Collagen (%)	BMMC (*n* = 8)	50.50	±6.61	
	Nanoparticle (*n* = 8)	44.74	±9.69	
	(BMMC) +NP (*n* = 8)	46.50	±9.81	
Type III	Control (*n* = 10)	35.81	±8.45	<0.0001
Collagen (%)	BMMC (*n* = 8)	100.5	±8.07	
	Nanoparticle (*n* = 8)	115.7	±23.45	
	(BMMC) +NP (*n* = 8)	112.2±	±22.19	
Infarct area (%)	Control (*n* = 10)	273.2	±1.87	0.979
	BMMC (*n* = 8)	306.6	±2.41	
	Nanoparticle (*n* = 8)	367.8	±3.57	
	(BMMC) +NP (*n* = 8)	309.2	±2.79	

Data are expressed as mean ± SD. BMMC + NP, bone marrow mononuclear stem cells + nanoparticle; Statistical analysis was performed with one-way ANOVA followed by the Kruskal–Wallis test, Values of * *p* < 0.0001 denote statistical significance.

## 4. Discussion

In the present study, we investigated the role of autologous bone marrow mononuclear cells (BMMC)-conjugated anti-inflammatory nanoparticles for cardiac repair after MI. Treatment with BMMC is considered a therapeutic alternative for myocardial infarction due to its low cost and absence of complications, although various studies have used different sources of cells for the treatment of MI. The therapeutic use of these cells in cardiac regenerative medicine includes BMMC, Mesenchymal stem cells (MSCs), human embryonic stem cells (hESCs), pluripotent stem cells (PSCs), induced pluripotent stem cells (iPSCs), and those that have the capacity to differentiate into all cell types. [13,14].

In addition, studies have suggested numerous limitations of cell transplantation, such as scarce retention of stem cells, grafts with low cell density, and excessive cell death in the infarction area [15]. Our findings demonstrated that BMMC combined with anti-inflammatory nanoparticles result in great potential for regenerative medicine. Many results have indicated that cell injection into a myocardial infarction model has resulted in some improvement in cardiac function [16,17]. We examined via an echocardiography study the effects of treatment four weeks post-implantation; some data have shown that BMMC combined with anti-inflammatory nanoparticles recover diastolic and systolic function, and improve the function ejection fraction in the BMMC+15d-PGJ2 group in the infarcted region.

In our previous study, it has been shown that the bone marrow mononuclear stem cells were functionally effective in the chronic myocardial infarction with ventricular dysfunction, which suggests improved myocardial function, especially for the mechanism of angiogenesis at the site of transplantation [10,18].

Regarding the reduction in scar area after combined treatment, MI size was significantly reduced in the BMMC+15d-PGJ2 group when measured by calculating from the microscopy images of slices of hearts stained with Gomori’s trichrome. Regarding indices of collagen type I in the infarcted area, there was also a significant decrease, which may contribute to cardiomyocyte survival after ischemia. Passino et al., 2015, have shown that excess and deregulated fibrosis may contribute to pathological myocardium remodeling by the local reduction of collagen I [19]. Other investigators have used animal models and suggest that treatment with inflammatory nanoparticles reduces myocardial infarction size and improves cardiac function after MI [20].

TGF-β and BMMC cells are likely to communicate bidirectionally to reduce cardiac hypertrophy in repair tissue by decreasing interstitial fibrosis in the infarcted heart. We believe that our TGF-β results provide a new view of the cellular mechanism in fibrous tissue deposition after myocardial infarction. However, the mechanisms responsible for these effects remain unknown. Studies suggest an important role for TGF-β in regulating hypertrophic cardiac remodeling and that TGF-β may be involved in the recruitment of mononuclear cells [21].

A major problem that we encounter today is related to the treatment of myocardial infarction with cardiogenic shock, where there is a high mortality rate [22,23,24]. Thus, the treatment of bone marrow cells with nanoparticles can be an alternative, as it may block the inflammatory effects of the infarction with the possibility of tissue regeneration.

This might indicate that BMMC combined with anti-inflammatory nanoparticles can enhance tissue perfusion, contribute to angiogenesis, and preserve or regenerate myocardial tissue by producing paracrine effects in situ. Our results from immunohistochemical analyses demonstrated the presence of angiogenesis identified in the transplanted region and that there was significant expression of connexin-43 in gap junctions, showing a more effective electrical and mechanical integration of the host myocardium.

Previous studies have shown that different types of stem cell transplantation facilitate permanence on-site and the formation of angiogenesis [25,26,27].

## 5. Conclusions

In conclusion, the present study demonstrated that a combination of BMMC and anti-inflammatory nanoparticles can be innovative, with the potential beneficial use in regenerative therapy. Thus, the application of nanoparticle technology in the prevention and treatment of MI is an emerging field and can be a strategy for cardiac repair.

## Figures and Tables

**Figure 1 jfb-13-00059-f001:**
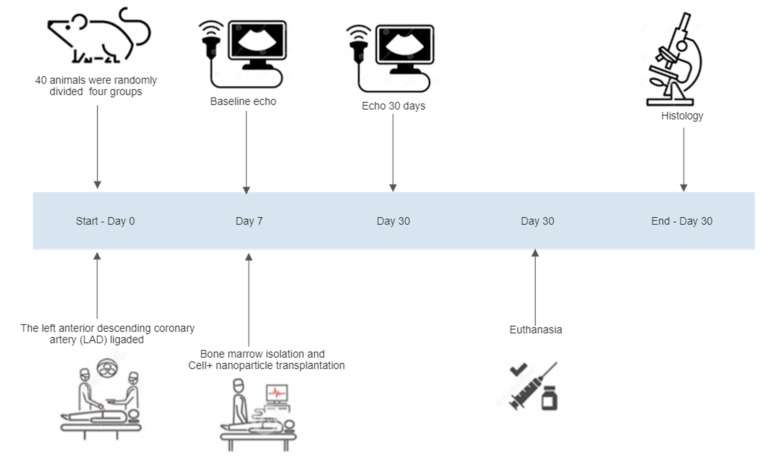
Experimental design: Randomization and divided group, bone marrow stem cell isolation and chronic myocardial infarction, baseline echo, the implant of cells + nanoparticle, echo 30 days, euthanasia, and histopathological analysis.

**Figure 2 jfb-13-00059-f002:**
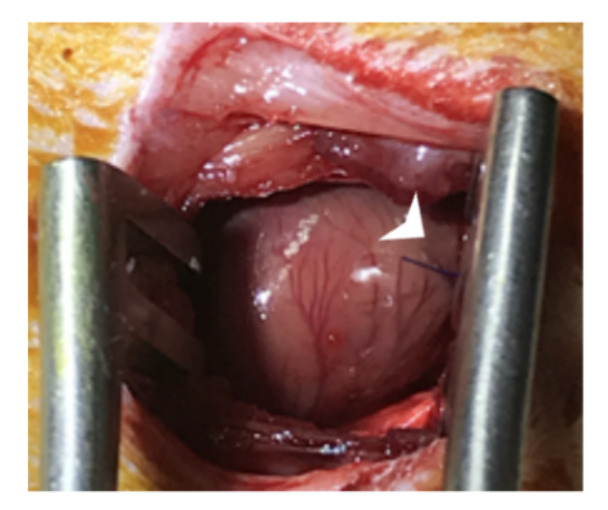
MI surgery was performed by ligation of the left anterior descending coronary artery (arrow) with a 7–0 prolene suture.

## Data Availability

Not applicable.

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
