# Peer review of "Autologous Bone Marrow Mononuclear Cells (BMMC)-Associated Anti-Inflammatory Nanoparticles for Cardiac Repair after Myocardial Infarction"

_jfb, 2022, doi:10.3390/jfb13020059_

Round 1

Reviewer 1 Report

This manuscript investigated the cotransplantation of bone marrow mononuclear cells with 15d-PGJ2 Nanoparticles for myocardial infarction treatment. The topic is of interest, but the study design, as well as the outputs of the data are not in proper formats, making the manuscript difficult to interpret. There are also lots of errors in the manuscript needed to be corrected. 
1. it would be better to claim that a chronic myocardial infarction model was employed for study, since the therapeutics were injected 7 days after modeling.

2. In figure 2, the authors claimed they used a silk suture for ligation. I doubt that this would lead to ischemia/reperfusion injury, as a result of the suture absorbable, but not the permanent infarction.

3. In the subtitle 3.1, minor error, BMNC.

4. Cardiac function measurement didn’t show significant effectiveness of cotransplantation versus BMMC or nanoparticles alone. 
5. It’s not scientific to show the data in table3 as percentage of positive cells. Calculations of integral density and staining area would be better to reflect the relative levels.

6. It’s swing that the n number in control group. Consistency should be checked out through the manuscript.

7.  In table 4, the quantitative data was shown as percentage, it’s hard to interpret the data.

Author Response

We thank the Referee for their interest in our work and for helpful comments that will greatly improve the manuscript and we have tried to do our best to respond to the points raised.

The Referee has brought up some good points and we appreciate the opportunity to clarify our research objectives and results. As indicated below, we have checked all the general and specific comments provided by the Referee and have made necessary changes accordingly to their indications.

Autologous Bone marrow mononuclear cells (BMMC)- Associated Anti-Inflammatory Nanoparticles for Cardiac Repair after Myocardial Infarction

Laercio Uemura1 and 1Rossana Baggio Simeoni1 *, Paulo André Bispo Machado Júnior 1, Gustavo Gavazzoni Blume1, Luize Kremer Gamba 1, Murilo Sgarbossa Tonial 1, Paulo Ricardo Baggio Simeoni1, Victoria Stadler Tasca Ribeiro1, Rodrigo Silvestre3, Katherine A. T. de Carvalho2, Marcelo Henrique Napimoga4; Júlio Cesar Francisco1 and Luiz Cesar Guarita-Souza 1

1 - Reviewer 1

  1. it would be better to claim that a chronic myocardial infarction model was employed for study, since the therapeutics were injected 7 days after modeling.

Response: Thanks for recommendation, a paragraph has claim been incorporated in text.

  1. In figure 2, the authors claimed they used a silk suture for ligation. I doubt that this would lead to ischemia/reperfusion injury, as a result of the suture absorbable, but not the permanent infarction.

Response: Thanks for commenting, the description of the figure suture will be corrected.

  1. In the subtitle 3.1, minor error, BMNC.

Response: Thanks for commenting, the use correct acronyms will be corrected.

  1. Cardiac function measurement did not show significant effectiveness of transplantation versus BMMC or nanoparticles alone.

  1. It’s not scientific to show the data in table3 as percentage of positive cells. Calculations of integral density and staining area would be better to reflect the relative levels.

Response: Thanks for commenting, the description of the data in table 3 will be corrected.

  1. It’s swing that the n number in control group. Consistency should be checked out through the manuscript.

Response: Thank you for commenting, but the questions the number in control group will be corrected.

  1.  In table 4, the quantitative data was shown as percentage, it’s hard to interpret the data.

Response: Thank you for commenting, but the questions the number in control group will be corrected.

Reviewer 2 Report

The manuscript by Baggio Simeoni et al proposes a combination therapy of bone marrow-derived mesenchymal stem cells and the 15-Deoxy-∆-12,14-Prostaglandin J2 (15d-PGJ2) nanoparticle for treatment of acute myocardial infarction. Within the understandable limitations of the rat animal model employed, they provide interesting evidence that this approach could be considered for a better outcome following this dangerous cardiac event.

The central idea of the manuscript is novel. Although this specific nanoparticle has been used in other studies and proposed to have antitumoural, anti-inflammatory, antioxidative, antifibrotic, and antiangiogenic effects, its action in the context of acute myocardial infarction had not been analyzed.

The manuscript is sound, but it requires language/grammar editing and its structure could be improved per the following notes:

Lines 72-75: Experimental group names should be simplified and used consistently throughout the manuscript. Smaller acronyms could be used, for example: Control, NP -Nanoparticle, bmMSCs - bone marrow-derived Mesenchymal stem cells only  and bmMSCs+NP -  bone marrow-derived Mesenchymal stem cells with nanoparticle.

Line 87: The centrifugation conditions should be converted from rpm to force, expressed in g.

Figure 2 seems unnecessary to this reviewer. If the authors feel strongly that this figure needs to be included, moving it to supplementary data should be considered.

Figure 4: The misaligned time points for pre and post AMI are confusing. If the duration of the treatment was the same for all conditions, all treatments should be aligned, beginning at pre-AMI and finish at post-AMI. Legend is too small and in portuguese. Font size shall be increased and words translated to english. Per common manuscript organization practices, the table below it should be separated as it own results entity and appropriately labeled.

Tables 1 and 2 - Experimental group nomenclature should be consistent with what is specified in lines 72-75 and to each other. In table one the nanoparticle is called out as both Nanoparticle and 15pgj. In table 2 it is called simply NP.

Table 2 is hard to follow, the data should be presented as bar graphs as well, graphically highlighting significant differences between groups where appropriate, as it was done for Table 4 and Figure 6.

Table 3 should be moved to after figure 5, and similar to table 2, this data would benefit from being presented as bar graphs as well, highlighting significant differences between groups where appropriate, as it was done for Table 4 and Figure 6. If one-way ANOVA was used throughout the analysis was used for all the markers, the test column can be removed from the table and the test can be specified in the table caption. If other tests were used for other markers they shall be specified.

Figure 5: Desmin is misspelled in the figure legend

Table 4 should be moved to after figure 6.

Line 319 - the word reduced is redundant as should be eliminated

Author Response

We thank the Referee for their interest in our work and for helpful comments that will greatly improve the manuscript and we have tried to do our best to respond to the points raised.

The Referee has brought up some good points and we appreciate the opportunity to clarify our research objectives and results. As indicated below, we have checked all the general and specific comments provided by the Referee and have made necessary changes accordingly to their indications.

Autologous Bone marrow mononuclear cells (BMMC)- Associated Anti-Inflammatory Nanoparticles for Cardiac Repair after Myocardial Infarction

Laercio Uemura1 and 1Rossana Baggio Simeoni1 *, Paulo André Bispo Machado Júnior 1, Gustavo Gavazzoni Blume1, Luize Kremer Gamba 1, Murilo Sgarbossa Tonial 1, Paulo Ricardo Baggio Simeoni1, Victoria Stadler Tasca Ribeiro1, Rodrigo Silvestre3, Katherine A. T. de Carvalho2, Marcelo Henrique Napimoga4; Júlio Cesar Francisco1 and Luiz Cesar Guarita-Souza 1

Lines 72-75: Experimental group names should be simplified and used consistently throughout the manuscript. Smaller acronyms could be used, for example: Control, NP -Nanoparticle, bmMSCs - bone marrow-derived Mesenchymal stem cells only and bmMSCs+NP -  bone marrow-derived Mesenchymal stem cells with nanoparticle.

Response: Thanks for commenting, the use correct acronyms will be corrected.

Line 87: The centrifugation conditions should be converted from rpm to force, expressed in g.

Response: Thanks for commenting, the use correct rpm to G force will be corrected.

Figure 2 seems unnecessary to this reviewer. If the authors feel strongly that this figure needs to be included, moving it to supplementary data should be considered.

Figure 4: The misaligned time points for pre and post AMI are confusing. If the duration of the treatment was the same for all conditions, all treatments should be aligned, beginning at pre-AMI and finish at post-AMI. Legend is too small and in portuguese. Font size shall be increased and words translated to english. Per common manuscript organization practices, the table below it should be separated as it own results entity and appropriately labeled.

Response: Thanks for commenting, the figure 4 has been corrected, displayed on a bar chart.

Tables 1 and 2 - Experimental group nomenclature should be consistent with what is specified in lines 72-75 and to each other. In table one the nanoparticle is called out as both Nanoparticle and 15pgj. In table 2 it is called simply NP.

Response: Thanks for commenting, the abbreviation has been corrected.

Table 2 is hard to follow, the data should be presented as bar graphs as well, graphically highlighting significant differences between groups where appropriate, as it was done for Table 4 and Figure 6.

Response: Thanks for commenting, the table 4 has been moved after figure 6 and the test column will be removed.

Table 3 should be moved to after figure 5, and similar to table 2, this data would benefit from being presented as bar graphs as well, highlighting significant differences between groups where appropriate, as it was done for Table 4 and Figure 6. If one-way ANOVA was used throughout the analysis was used for all the markers, the test column can be removed from the table and the test can be specified in the table caption. If other tests were used for other markers they shall be specified.

Response: Thanks for commenting, the table 2 has been moved after figure 5 and the test column will be removed.

Figure 5: Desmin is misspelled in the figure legend

Response: Thanks for commenting, the use correct description will be corrected.

Table 4 should be moved to after figure 6.

Response: Thanks for commenting, the table has been moved after figure 6.

Line 319 - the word reduced is redundant as should be eliminated

Answer: Thanks for commenting, the paragraph has been deleted.

Reviewer 3 Report

I have reviewed the manuscript entitled 'Autologous Bone marrow mononuclear cells (BMMC)- Associ- 2 ated Anti-Inflammatory Nanoparticles for Cardiac Repair after 3 Myocardial Infarction'
The investigation appear to be very interesting and could present promising changes in the current treatment strategies. 
The authors should test the ntitled 'Autologous Bone marrow mononuclear cells (BMMC)- Associ- 2 ated Anti-Inflammatory Nanoparticlesin high risk patients such as cardiogenic shock in order to test the endpoints clearly.
The high mortality of cardiogenic shock patients should be used to test the effect of this treatment. The potential future studies should be discussed, please consider citing 'Predictors of In-Hospital Mortality in Patients With ST-Segment Elevation Myocardial Infarction Complicated With Cardiogenic Shock'

Author Response

Dear Reviewer, we include your suggestion in the discussion, with reference to
your citation.
A major problem that we encounter today is related to the treatment of
myocardial infarction with cardiogenic shock, where there is a high mortality
rate. Thus, the treatment of bone marrow cells with nanoparticles can be an
alterantive, as it may block the inflammatory effects of the infarction with the
perspctive of tissue regeneration.

Ref. Hayıroğlu Mİ, Keskin M, Uzun AO, Yıldırım Dİ, Kaya A, Çinier G, Bozbeyoğlu E, Yıldırımtürk Ö, Kozan Ö, Pehlivanoğlu S. Predictors of In-Hospital Mortality in Patients With ST-Segment Elevation Myocardial Infarction Complicated With Cardiogenic Shock. Heart Lung Circ. 2019 Feb;28(2):237-244. doi: 10.1016/j.hlc.2017.10.023. Epub 2017 Nov 14. PMID: 29191504.

Round 2

Reviewer 1 Report

All my questions have been addressed. I have no further questions.